# Identification and Functional Analysis of *KH* Family Genes Associated with Salt Stress in Rice

**DOI:** 10.3390/ijms25115950

**Published:** 2024-05-29

**Authors:** Qinyu Xie, Yutong Zhang, Mingming Wu, Youheng Chen, Yingwei Wang, Qinzong Zeng, Yuliang Han, Siqi Zhang, Juncheng Zhang, Tao Chen, Maohong Cai

**Affiliations:** 1Zhejiang Provincial Key Laboratory for Genetic Improvement and Quality Control of Medicinal Plants, College of Life and Environmental Science, Hangzhou Normal University, Hangzhou 311121, China; 2Institute of Crop and Nuclear Technology Utilization, Zhejiang Academy of Agricultural Sciences, Hangzhou 310021, China

**Keywords:** *KH* family, salt stress, rice, genome-wide identification

## Abstract

Salinity stress has a great impact on crop growth and productivity and is one of the major factors responsible for crop yield losses. The K-homologous (KH) family proteins play vital roles in regulating plant development and responding to abiotic stress in plants. However, the systematic characterization of the *KH* family in rice is still lacking. In this study, we performed genome-wide identification and functional analysis of *KH* family genes and identified a total of 31 *KH* genes in rice. According to the homologs of *KH* genes in *Arabidopsis thaliana*, we constructed a phylogenetic tree with 61 *KH* genes containing 31 *KH* genes in *Oryza sativa* and 30 *KH* genes in *Arabidopsis thaliana* and separated them into three major groups. In silico tissue expression analysis showed that the *OsKH* genes are constitutively expressed. The qRT-PCR results revealed that eight *OsKH* genes responded strongly to salt stresses, and *OsKH12* exhibited the strongest decrease in expression level, which was selected for further study. We generated the *Oskh12-knockout* mutant via the CRISPR/Cas9 genome-editing method. Further stress treatment and biochemical assays confirmed that *Oskh12* mutant was more salt-sensitive than Nip and the expression of several key salt-tolerant genes in *Oskh12* was significantly reduced. Taken together, our results shed light on the understanding of the *KH* family and provide a theoretical basis for future abiotic stress studies in rice.

## 1. Introduction

The K-homologous (KH) family proteins belong to DNA/RNA-binding proteins containing the conserved KH domain. The KH domain was first identified in human heterogeneous nuclear ribonucleoprotein K (hnRNPK) [1], which consists of approximately 70 amino acids that can be folded into two completely different ways, named type Ⅰ and type Ⅱ folding, respectively [2,3]. The *KH* family genes play important biological functions in animals. For example, HRPK-1, a KH domain protein, controls Caenorhabditis elegance development by affecting miRNA processing and enhancing miRNA/miRISC gene regulatory activity [4]. KHSPR (KH-type splicing regulatory protein) provided a potential link with human diseases [5]. SAM68 (also known as KH-domain-containing, RNA-binding, signal-transduction-associated 1 (KHDRBS1)) plays vital roles in human cancers including lung adenocarcinoma [6], hepatic gluconeogenesis [7], and breast cancer [8].

In addition to their important functions in animals, *KH* family genes also play an important role in plants, which have been shown to participate in plant growth, development, and hormone-related processes, and adapt to environmental stresses. Several members of the *KH* family were reported to be involved in flowering time. For instance, *FLK*, *GRP7*, *GRP8*, *KHZ1*, and *KHZ2* function as flowering promoters, while *PEPPER*, *FLY*, and *HEN4* function as flowering suppressors in *Arabidopsis thaliana* [9,10,11,12,13,14,15,16]. In rice, SPIN1, a KH protein ubiquitinated by the E3 ubiquitin ligase SPL11, delays flowering independently of day length [17]. Recent studies have shown that *AtKH9* (*At2g38610*) and *AtKH29 (At5g56140*) can respond to abscisic acid (ABA) and salicylic acid (SA) in *Arabidopsis thaliana* [18]. ABA plays an important role in plant adaptation to abiotic stress. KH proteins are involved in the process of abiotic stress in plants. KH domain protein REGULATOR OF CBF GENE EXPRESSION 3 (RCF3) negatively regulates heat stress by impacting the expression of heat stress factors in *Arabidopsis thaliana* [19]. HOS5 (High Osmotic Stress Gene Expression 5) and AtSIEK (stress-induced protein with EXD1-like domain and KH domain) are both K homology (KH)-domain RNA-binding proteins that interact with FRY2/CPL1 whose mutants exhibit both ABA and salt-sensitive phenotypes [20,21,22,23]. The *KH* family genes have been widely identified and studied in *Arabidopsis*. However, the functional analysis of the *KH* family genes in rice was seldom reported.

Rice (*Oryza sativa*) is one of the most important crops in the world, and salinity stress is rated as one of the major abiotic stresses [24]. Soil saline–alkalization is a major abiotic stressor on the world’s agriculture, causing considerable damage to crop growth and resulting in serious losses in crop production [25,26,27]. The salt overly sensitive (SOS) signaling pathway plays a central role in exporting Na^+^ from the cytosol to the apoplast and conferring salt tolerance in plants [28,29,30,31,32]. Moreover, the salt tolerance of plants is also regulated by hormones such as ABA. ABA, as a signal molecule, regulates salt tolerance by mediating the expression of salt stress genes. Many genes regulating salt stress tolerance in rice have been identified. *SOS1*, *AKT1*, *NHX2*, *HKT2;1*, and *SKC1*(*HKT1;5*) are key regulators that positively modulate rice salt tolerance by maintaining homeostasis of K^+^/Na^+^ [29,33]. The core of ABA regulating plant salt tolerance is the activation and transcriptional regulation of SnRK2 protein kinase on multiple downstream target genes, thereby directly or indirectly regulating plant salt tolerance [34,35]. For example, the rice zinc finger protein OsbZIP23, containing ABRE elements in the promoter region, increases the susceptibility to salt and ABA in rice [36]. *GmSIN1* (*SALT INDUCED NAC1*) is induced by ABA and promotes root growth and salt tolerance in *Glycine max*, and the GmSIN1 transcription factor directly binds to the SIN1BM element on the promoters of salt tolerance genes, thereby regulating the response to salt stress [37]. Despite numerous genes that have been identified regulating salt stress in rice, the function of *KH* family genes in responding to salt stress remains unknown.

In this study, we systematically performed genome-wide identification and analysis of the *KH* family genes in rice, which bridges the research gap for *KH* gene family studies in rice. A total of 31 *KH* genes were identified, which were uniformly distributed in 12 chromosomes. The phylogeny and evolutionary relationships, conserved motifs, gene expression, and cis-element of *KH* family genes were comprehensively performed. The gene expression profiling revealed that all *OsKH* genes are constitutively expressed in various rice tissues. Furthermore, qRT-PCR analysis showed that the expressions of *OsKH1*, *OsKH10*, *OsKH12*, *OsKH13*, *OsKH18*, *OsKH27*, and *OsKH29* are significantly affected under salt stress. To elucidate the function of *OsKH12* in salt responses, with the highest decrease in expression level, we generated the *Oskh12* CRISPR knockout mutant. Further molecular experiments confirmed that *Oskh12* decreased tolerance to salt stress and the expression of several key salt-tolerant genes in *Oskh12* was significantly reduced, indicating that the *Oskh12* mutant was more salt-sensitive than Nip. In conclusion, this study systematically analyzed the response of the *KH* family genes in response to salt stress in rice for the first time and provided new genetic resources for rice breeding.

## 2. Results

### 2.1. Identification and Chromosomal Location of KH Family Genes in Rice

To identify the *KH* family members in rice, the CDSs of *KH* genes in rice were downloaded from the Phytozome website (https://phytozome-next.jgi.doe.gov/, accessed on 4 January 2024). A total of 31 *KH* family genes were finally screened and identified in rice. To understand the distribution characteristics of *OsKH* genes on chromosomes, the chromosomal locations of *OsKH* genes were analyzed by TBtools and results revealed that the 31 *OsKH* genes, not the gene clusters, were distributed on almost all twelve chromosomes, except for chromosome 11, and chromosome 1 has the most *OsKH* genes with seven (Figure 1A). These *KH* genes were named according to their position on the chromosome, in the order of *OsKH1* to *OsKH31*. Gene duplication analysis predicted five *OsKH* gene pairs in the rice genome, making a total of ten duplicated genes, including *OsKH6/OsKH19*, *OsKH7/OsKH18*, *OsKH9/OsKH21*, *OsKH15/OsKH31*, and *OsKH16/OsKH22* (Figure 1B). The duplication of rice *KH* genes was analyzed by the Advanced Circos of TBtools software v2.096. Amino acid sequence alignment showed that the KH domain of these genes has the highest similarity (Appendix A). These results demonstrated that several *KH* genes may appear in the course of gene duplication, and the segmental duplication events may be responsible for the expansion of *KH* genes in rice.

To explore the potential functions of the *KH* family genes, the CDSs of identified *KH* genes were downloaded from the Tair website (https://www.arabidopsis.org/, accessed on 4 January 2024). The comprehensive information of 31 *OsKH* genes and 30 *AtKH* genes was analyzed and is summarized in Table 1, including the number of amino acids, molecular weight (MW), theoretical PI, instability index, aliphatic index, the grand average of hydropathicity, and the subcellular location. These protein characteristics were analyzed through the Protein Paramter Calc of TBtools software v2.096 and the protein subcellular localization prediction of *KH* family proteins was analyzed by the PSORT website (https://wolfpsort.hgc.jp/, accessed on 4 January 2024). Among them, *AtKH30* encoded the largest protein, containing 866 amino acids, while *OsKH3* encoded the smallest protein, consisting of 133 amino acids. The protein molecular weight (MW) and isoelectric point (PI) of these proteins range from 14.7 (*OsKH3*) to 94.7 kDa (*AtKH30*), and 4.16 (*AtKH5*) to 10.07 (*OsKH3*), respectively. According to the analysis of the grand average of hydropathicity, the *KH* family proteins are generally hydrophilic (Table 1). To analyze subcellular localization, all KH proteins were predicted by the PSORT website. The results showed that most KH proteins (50.82%) are located in the nucleus, which is consistent with the localization characteristics of the RNA-binding protein. In addition, 24.6% of proteins are located in the cytoplasm, 14.75% of proteins are likely to be located in the chloroplast, and small partial proteins are separately located in the mitochondria (AtKH7 and OsKH2), cytoskeleton (OsKH5 and OsKH12), and peroxisome (AtKH5), which implies that the KH proteins may also play a function in chloroplast or mitochondria (Table 1).

### 2.2. Phylogenetic and Synteny Analysis of KH Genes in Arabidopsis thaliana and Oryza sativa

To investigate the genetic phylogenetic relationship of the *KH* family genes, the protein sequences of *Arabidopsis thaliana* (30 members) and *Oryza sativa* (31 members) were downloaded from the Phytozome website (https://phytozome-next.jgi.doe.gov/, accessed on 4 January 2024) and the Tair website (https://www.arabidopsis.org/, accessed on 4 January 2024), respectively, and were used to construct phylogenetic trees by MEGA7 using the neighbor-joining method. According to the phylogenetic tree, the 61 *KH* genes were classified into three groups: Group 1, Group 2, and Group 3. Group 1 consisted of the largest number of *KH* members, with 28 genes, followed by Group 2 with 23 genes, while Group 3 had the smallest with only 10 genes. Group 1 contained 13 and 15 *KH* genes in rice and *Arabidopsis*, respectively; Group 2 contained 12 and 11 *KH* genes in rice and *Arabidopsis*, respectively; and Group 3 contained 6 and 4 *KH* genes in rice and *Arabidopsis*, respectively (Figure 2A). Interestingly, the ratio of *OsKH* to *AtKH* in each group was almost 1:1, indicating that the *KH* family genes are conserved between the two species.

To further analyze the evolutionary relationship between *KH* genes of rice and Arabidopsis, we performed a comparative synteny analysis between *Arabidopsis thaliana* and *Oryza sativa* using the Dual Systeny Plot of TBtools software v2.096 (Figure 2B). There was a linear relationship in five pairs of genes, including *AtKH3*/*OsKH25*, *AtKH9*/O*sKH6*, *AtKH11*/*OsKH6*, *AtKH11*/*OsKH19*, and *AtKH29*/*OsKH11*, between the two species. The coding sequences of the five gene pairs are 850 to 2290 bp (Figure 2B and Table 1). This finding indicated that the *KH* genes were highly conserved in evolutionary terms.

### 2.3. Conserved Motif and Domain Analysis of the KH Family Proteins

To explore the potential functions of *KH* family proteins in detail, the conserved motif and domain of 61 KH proteins were analyzed using the MEME website (Figure 3). A total of 10 conserved motifs were identified and the detail motifs are shown in Appendix A. Notably, the three groups had significantly different compositions of motifs. The proteins of Group 1 contains ten motifs and has the largest number of motifs, and almost all genes in Group 1 contain motif 2, motif 5, motif 6, and motif 7. However, Group 2 mainly contained three motifs and most members of Group 3 contained two motifs (Figure 3A). Interestingly, motif 1 was found in almost all KH proteins in rice and *Arabidopsis*, while motif 3 and motif 4 were only found in the KH proteins of Group 2.

Conserved domain information was calculated by Batch CD-Search on the NCBI website and then visualized by the Gene Structure View of TBtools software v2.096. Conserved domain analysis revealed that nine domains exist in the *KH* family and the KH domain was found in entire KH members. In addition, Group 1 members contain the most KH domains, from two to five, and Group 2 members mainly contain both one KH domain and one other domain, such as STAR, ZnF_C3H1, and zf-CCCH (Figure 3B). Furthermore, the members of Group 3 included the KH domains, and most of them had only one KH domain (Figure 3B).

### 2.4. Tissues Expression Analysis of KH Members in Oryza sativa

The tissue-specific expression patterns of genes reflect their action sites and mechanisms. To analyze the expression profiles of *KH* genes in different tissues of rice, we screened the expression levels of *OsKH* genes in 12 rice tissues, including leaf blade, leaf sheath, root, stem, inflorescence, anther, pistil, lemma, palea, ovary, embryo, and endosperm, which were downloaded from the rice public database of RiceXPro (https://ricexpro.dna.affrc.go.jp/, accessed on 6 January 2024) and then visualized by the HeatMap of TBtools software v2.096. According to the expression results, *KH* genes showed slight differences in tissue expression. In general, most of the *OsKH* genes such as *OsKH4*, *OsKH9*, *OsKH12*, *OsKH13*, and *OsKH25* are constitutively expressed in these twelve tissues in rice. However, there are also a small number of *OsKH* genes that are not constitutively expressed. For example, the expressions of *OsKH23* in inflorescence, anther, and pistil are significantly higher than other tissues of the twelve tissues (Figure 4). Specifically, *OsKH4*, *OsKH9*, *OsKH10*, *OsKH12*, *OsKH13*, and *OsKH25* were evolutionarily close and had high expression levels in all rice tissues. On the contrary, *OsKH23*, *OsKH26*, *OsKH27*, and *OsKH28* showed low expression levels in the twelve tissues of rice (Figure 4). These results show that the genes that are more evolutionarily closely related have similar expression patterns. In brief, these data indicated that the *KH* genes may play a key role in the formation and development of different tissues with varied expression levels in rice.

### 2.5. Cis-Acting Element Analysis of KH Gene Promoters

To explore the potential functions of *KH* family genes, the 2000 bp upstream sequences of 61 *KH* genes in *Oryza sativa* and *Arabidopsis thaliana* were extracted from the NCBI website (https://www.ncbi.nlm.nih.gov/, accessed on 6 January 2024) to analyze the potential cis-elements using the PlantCARE (https://bioinformatics.psb.ugent.be/webtools/plantcare/html/, accessed on 6 January 2024) website and visualized by the Basic Biosequence View of TBtools software v2.096. The promoter regions of *KH* genes were analyzed (Figure 5), and six main responsive elements were identified, related to ABA (abscisic acid), GA (gibberellin), drought stress, endosperm, meristem, and light responsiveness. Among them, GA and endosperm expression elements are specific to the *KH* genes of *Oryza sativa* in these two species, while drought-inducibility and meristem expression elements are unique to the *KH* genes of *Arabidopsis thaliana* compared to rice (Figure 5A). In rice, there are 9 ABA responsiveness elements, 18 GA-responsive elements, and 5 endosperm expression elements (Figure 5B). In *Arabidopsis thaliana*, there are 45 ABA responsiveness elements, 20 drought-inducibility elements, 10 meristem expression elements, 137 light responsiveness elements, and no GA-responsive and endosperm expression elements (Figure 5C). These results suggest that *KH* family genes are involved in plant growth and development, and play an important role in plant response to hormone and abiotic stress response.

### 2.6. Expression Pattern Analysis of OsKHs under Salt Stress and ABA Treatment

Cis-acting element analysis of *KH* genes showed that the *KH* family genes mainly responded to light and ABA, and the ABA signal was closely related to salt stress. So, the response function of the *KH* genes to salt stress was mainly studied. To study the possible function of *OsKHs* in response to salt stress, the transcript level of 31 *OsKH* genes under the treatment of 300 mM NaCl was analyzed by qRT-PCR. After treatment with 300 mM NaCl for 0 h, 3 h, 6 h, and 12 h, the seedlings were collected for RNA extraction and qRT-PCR analysis. The expression analysis shows that the relative expression levels of *OsKH* genes in the leaf are more consistent with the tissue expression pattern. The expressions of *OsKH4* and *OsKH13* in rice leaves are relatively higher, which is consistent with the data in the database of Figure 4. Furthermore, the expressions of *OsKH23* and *OsKH28* in rice leaves were so low that they could not be detected by qRT-PCR, which further illustrates the authenticity of our quantitative results (Figure 6). The results found that the eight *KH* genes were strongly repressed from 3 h to 12 h, including *OsKH1*, *OsKH7*, *OsKH10*, *OsKH12*, *OsKH13*, *OsKH18*, *OsKH27*, and *OsKH29*. Among these genes, *OsKH12* exhibited the strongest decrease in expression level. Three *KH* genes, *OsKH2*, *OsKH8*, and *OsKH16* were significantly upregulated at 6 h (Figure 6). Following ABA treatment, the expressions of *OsKH5*, *OsKH13*, *OsKH20*, *OsKH27*, and *OsKH30* were considerably down-regulated at all three time points (Figure 7). Primers used in this assay are listed in Appendix A. These findings indicated these genes are possibly related to salt or ABA signaling pathways. 

### 2.7. The Oskh12-Knockout Mutant Decreased Tolerance to Salt Stress in Rice

The expression pattern of *KH* genes under salt stress showed that *OsKH12* responded to salt stress to a large extent, which could be used as a representative of the *KH* gene family in rice. To elucidate the function of *KH* genes in response to salt stress, the expression of *OsKH12*, which was significantly reduced under the treatment of NaCl, was selected to be knocked out in rice using a CRISPR/Cas9 genome-editing approach. Two-week rice seedlings of Nip and homozygous *Oskh12* mutant were treated with 150 mM NaCl. The phenotype showed that the *Oskh12* mutant exhibited earlier wilted phenotypes and a lower survival rate than Nip (Figure 8A,B). These data confirmed that *Oskh12* was more salt-sensitive than Nip. In addition, ROS levels can reflect the extent of damage to the plant. Under salt stress, the damage degree of plants with strong salt tolerance will be lower. ROS staining by NBT displayed that the ROS level of *Oskh12* was significantly higher than Nip, which is consistent with the sensitive phenotype of *Oskh12*. The ROS level of *Oskh12* increased to a greater extent than Nip under the 300 mM NaCl treatment compared to the control (Figure 8C,D). These results illustrated that the tolerance of salt stress was decreased in *Oskh12*.

To investigate the effect of *OsKH12* on salt-related genes, the nine key salt-responsive genes (*LEA3*, *DREB1A*, *SKC1*, *HKT2;1*, *AKT1*, *iSAP8*, *NHX2*, *SOS2*, and *DSR2*) were selected for expression analysis under the treatment of NaCl. The expression of *LEA3*, *DREB1A*, *SKC1*, *HKT2;1*, *AKT1*, *iSAP8*, *NHX2*, *SOS2*, and *DSR2* were all induced by salt stress [38,39]. Overexpression of *iSAP8* and *DSR2* can enhance the tolerance to salt and drought in rice [40,41]. Also, *DREB1A* has been discovered to enhance salt tolerance both in *Oryza sativa* and *Arabidopsis thaliana* [42]. *SKC1*, *HKT2;1*, *AKT1*, *NHX2,* and *SOS2* were induced by salt stress in rice [43,44,45,46]. The qRT-PCR results showed that, under the treatment of 300 mM NaCl, the expression of *LEA3*, *DREB1A*, *SKC1*, *HKT2;1*, and *iSAP8* was significantly down-regulated in *Oskh12* compared to Nip, which was consistent with the phenotype of *Oskh12* under the treatment of NaCl (Figure 9). For instance, the expression level of *LEA3* was significantly increased in both Nip and *Oskh12* under salt treatment conditions. However, the upregulation of *LEA3* was markedly lower in *Oskh12* compared to Nip (Figure 9). The expression level of *SKC1* was significantly upregulated after salt treatment in Nip; in contrast, the expression level of *SKC1* did not show a significant change in *Oskh12* under salt treatment (Figure 9). Another four genes, including *AKT1*, *NHX2*, *SOS2*, and *DSR2* showed no significant differences in expression levels in *Oskh12* (Figure 9). Primers used in this assay are listed in Appendix A. These results further confirmed that *OsKH12* participates in the regulation of salt response and regulates the salt stress response of plants by regulating the expression of salt-responsive genes. 

### 2.8. OsKH12 Is Related to the Proteins Containing the RRM Domain

To further identify the involvement of OsKH12 in the salt stress response, we predicted the proteins to which OsKH12 may bind by the STRING website (https://cn.string-db.org/, accessed on 11 January 2024). The predictions showed the three proteins that OsKH12 was most likely to bind to in rice and the gene IDs encoding these three proteins are Os07g0138700, Os02g0122800, and Os01g0866600 in rice (Figure 10). Among the three proteins, two proteins contain the RRM domain, which is the main RNA-binding domain involved in pre-mRNA splicing (Figure 10).

## 3. Discussion

The K-homologous (KH) protein is a type of nucleic acid-binding protein containing the KH domain, which plays an important role in the development and abiotic stress response [1,19,47,48]. However, systematic characterization and functions of the *KH* family in rice are still absent. In this study, we performed genome-wide identification of *KH* family genes in rice and Arabidopsis, and a total of 31 and 30 *KH* genes were identified, respectively (Figure 1). The comprehensive information was further analyzed including duplication analysis, protein sequence length, MW and pI, instability index, aliphatic index, grand average of hydropathicity, and subcellular localization. The phylogenetic and synteny analysis indicated that the evolution of *KH* gene families in the two plants was relatively conservative (Figure 2). We also performed conserved domain analysis of KH proteins and tissue expression of *KH* family genes to explore the potential function of the *KH* family, which revealed that all KH proteins contain the KH domain and the *OsKH* genes expressed in all rice tissues (Figure 3 and Figure 4).

To investigate the potential function of *OsKHs* in response to salt stress, the expression level of 31 *OsKH* genes under salt and ABA stress was analyzed by qRT-PCR. We found several key genes that showed obvious changes in response to abiotic stress (Figure 6 and Figure 7). According to the results of qRT-PCR, *OsKH12* was selected to investigate the function in response to salt stress including the phenotype analysis and expression of salt-responsive marker genes in *Oskh12-crispr* mutant. Our results confirmed that OsKH12 participates in the regulation of salt response by regulating the expression of salt-responsive genes (Figure 8 and Figure 9). The salt tolerance phenotypes of *KH* genes were also reported in other plants, such as *AtSIEK* [23] and *FRY2/CPL1* [21,49]. Moreover, in response to abiotic stress, *KH* genes also play vital roles in the regulation of plant growth and development. HEN4 containing the KH domain delays flowering time by upregulation of the *FLC* and *MAF4*, which are the MADS-box repressor genes in *Arabidopsis thaliana* [50]. SPIN1 delays the heading date by downregulating the flowering promoter gene *Hd3a* (*Heading date 3a*) in short days and by targeting Hd1-independent factors in long days in *Oryza sativa* [17]. These findings provide clues that the *OsKH* genes may be involved in the regulation of plant growth and development, such as heading date in rice.

According to the protein properties of *KH* family proteins, KH proteins function as nucleic acid-binding proteins and mainly regulate various physiological processes by regulating the expression of other genes through post-transcriptional regulation. The KH proteins do not directly regulate the physiological phenotype of plants. To predict the partner proteins of the *KH* family, we analyzed the possible interaction proteins of OsKH12 and found that the OsKH12 protein is likely to interact with proteins containing the RRM domain (Figure 10). The RRM domain is a type of RNA-binding protein, like the KH domain. There is a study that shows that the plant-specific splicing factor OsRS33 (Os02g0122800) plays a crucial role during plant responses to abiotic stresses including salt stress [51]. Consequently, it is reasonable to speculate that KH proteins can interact with many RNA-binding proteins to regulate abiotic stress response through pre-mRNA splicing in plants. The underlying molecular mechanism of KH proteins regulating physiological processes has great value for future research. In summary, these results suggested that *OsKH12* plays an important role in response to salt stress, which explains the significance of the *KH* genes for salt stress. Our study may provide better help to understand the source and function of *KH* family genes.

## 4. Materials and Methods

### 4.1. Identification of the KH Members in Oryza sativa

To perform genome-wide identification of the *KH* gene family, the whole genome of *Oryza sativa* and *Arabidopsis thaliana* and their annotation information were downloaded from Ensembl Plants (https://plants.ensembl.org/index.html, accessed on 3 January 2024) [52]. To find the *KH* family sequences in these two species, the hidden Markov model (HMM) with an E value < 10^−5^ of the KH domain (PF00013) [18] was downloaded from Pfam (http://pfam.xfam.org/, accessed on 3 January 2024) [53].

The *KH* family members coding sequences (CDSs) and peptide sequences of *Oryza sativa* and *Arabidopsis thaliana* were downloaded from the Phytozome website (https://phytozome-next.jgi.doe.gov/, accessed on 4 January 2024) and the Tair website (https://www.arabidopsis.org/, accessed on 4 January 2024), respectively. After removing duplicate transcripts, a total of 61 *KH* family genes were identified in the two species, named *AtKH1* to *AtKH30* and *OsKH1* to *OsKH31*. Basic information, such as the molecular weight of proteins, was analyzed by ExPASy (https://www.expasy.org/, accessed on 4 January 2024) [54]. The subcellular localization of *KH* family proteins was predicted using PSORT (https://www.genscript.com/psort.html, accessed on 4 January 2024).

### 4.2. Chromosomal Location and Duplication Analysis of KH Genes in Oryza sativa

To identify chromosomal locations, TBtools software v2.096 (https://github.com/CJChen/TBtools, accessed on 4 January 2024) was used to map *KH* genes on 12 chromosomes in *Oryza sativa* based on gene annotation information [55]. The data source is based on the Ensembl Plants database [52]. To further analyze gene replication, duplicated events were analyzed using the Advanced Circos of TBtools software v2.096.

### 4.3. Phylogenetic Tree and Collinearity Analysis of KH Proteins

To investigate the evolutionary relationship between *Oryza sativa* and *Arabidopsis thaliana*, the protein sequences of KH members were compared using the ClustalW program. The phylogenetic tree was constructed by MEGA7 using the neighbor-joining method. Tree nodes were evaluated through 1000 repeated bootstrap analyses. The final phylogenetic trees were modified using ITOL (https://itol.embl.de/, accessed on 5 January 2024) [56]. *Arabidopsis thaliana* and *Oryza sativa* were selected to analyze the collinearity of the *KH* family using the multicollinearity scan toolkit.in TBtools.

### 4.4. Conserved Motif and Domain Analysis of KH Family Proteins

The conserved motif of the full-length KH proteins was analyzed using the online MEME website (https://meme-suite.org/meme/, accessed on 5 January 2024) [57] by the zero or one occurrence per sequence (zoops) strategy and set the number of searchable motifs to 10. Conserved domain information of KH proteins was calculated by Batch CD-Search (https://www.ncbi.nlm.nih.gov/Structure/bwrpsb/bwrpsb.cgi, accessed on 5 January 2024) [58].

### 4.5. Expression Pattern Analysis of KH Genes in Oryza sativa

The normalized signal intensity data of *KH* genes in different tissues were downloaded from RiceXPro (https://ricexpro.dna.affrc.go.jp/, accessed on 6 January 2024) [59] to reveal the expression patterns of all *KH* family genes in *Oryza sativa*. Differential expression analysis and the heatmap were visualized using TBtools.

### 4.6. Cis-Regulatory Elements Analysis of KH Gene Promoters

The promoter sequences (2000 bp upstream of *KH* genes) of the *Arabidopsis thaliana* and *Oryza sativa* were downloaded from the NCBI website (https://www.ncbi.nlm.nih.gov/, accessed on 6 January 2024). These promoter sequences were used to predict cis-acting elements by the public website PlantCARE (https://bioinformatics.psb.ugent.be/webtools/plantcare/html/, accessed on 6 January 2024) [60].

### 4.7. Plant Materials and Methods of Stress Treatment

The Nipponbare was used for rice transformation to construct the *Oskh12* CRISPR mutant. The *Oskh12* mutant and Nip were used to analyze the responses to abiotic stresses in rice. Rice plants were grown at 32 ± 2 °C, in 1/2 Murashige and Skoog culture medium (MS), under LDs (long day, 14 h light/10 h dark). Two-week-old seedlings of rice were separately processed with 150 mM NaCl, 300 mM NaCl, and 50 µM ABA in 1/2 MS medium. For qRT-PCR analysis, the acute treatment method was adopted, with 300 mM NaCl and 50 µM ABA, and the treatment time was 24 h. Moreover, the phenotypic material was treated chronically, with 150 mM NaCl, and the treatment time was 3 days. After observing the phenotype, the seedlings were then collected for RNA extraction and qRT-PCR analysis. 

### 4.8. RNA Extraction and qRT-PCR Analysis

The leaves of two-week seedlings were used for the extraction of total RNA. The total RNA was extracted using the RNA-easy Isolation Reagent (Vazyme) and then synthesized cDNA with Evo M-MLV RT Mix Kit (Accurate Biology). qRT-PCR analyses were performed in the CFX384 Real-Time System (Bio-Rad) with SYBR Green Premix Pro Taq HS qPCR Kit (Accurate Biology). The *Oryza sativa Ubiquitin* (*UBQ*) gene was used as an internal control. Primers used in this assay are listed in Appendix A.

### 4.9. ROS Staining

ROS level was measured by nitro tetrazolium blue chloride (NBT). The higher the ROS level, the darker the color and the larger the area of the NBT stain. One-week-old seedlings of Nip and *Oskh12* were treated in 300 mM NaCl for 16 h before the seedling staining. The seedlings were incubated in 25 mM HEPES buffer (pH = 7.6) containing 1 mg/mL NBT (HWRK CHEM) for one hour at room temperature for dark staining and then destained in 95% ethanol. The root tips were then photographed using a light microscope. 

## Figures and Tables

**Figure 1 ijms-25-05950-f001:**
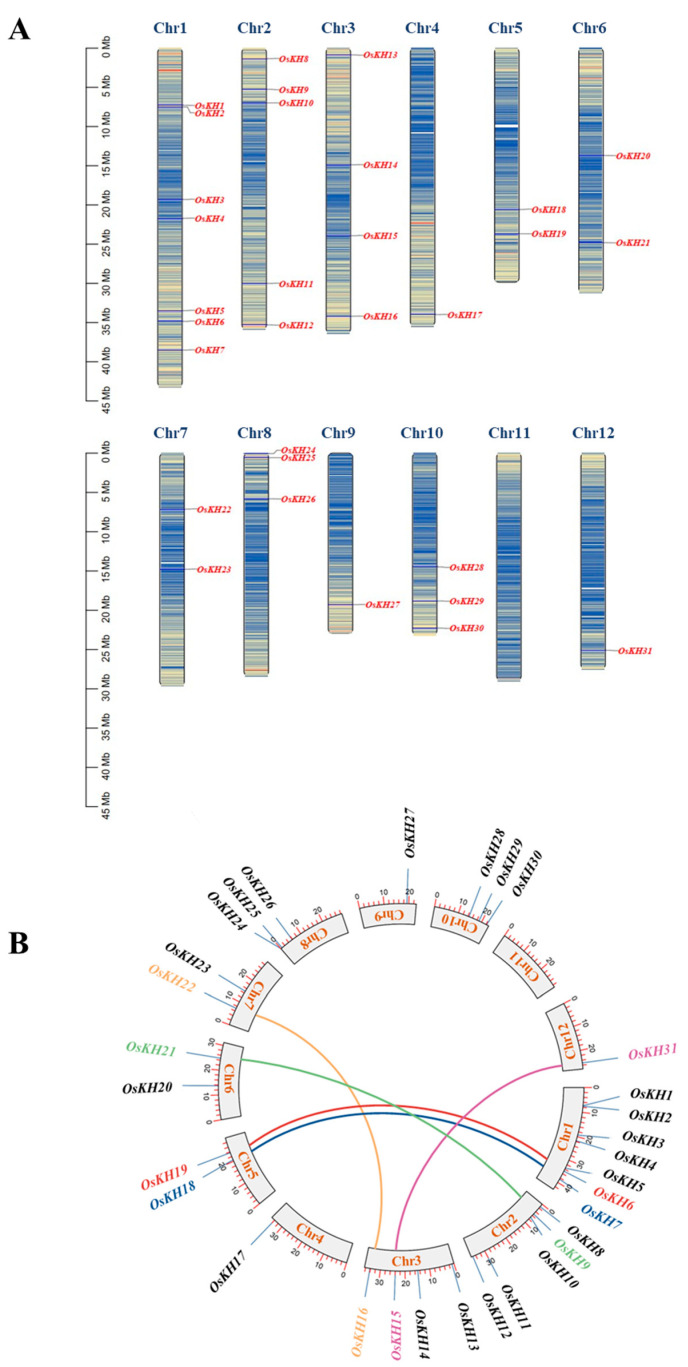
Chromosomal localization and collinearity analysis of 31 *KH* family genes in rice. (**A**) Chromosomal localization of thirty-one *KH* family genes in rice. The scale bars on the left indicate the length (Mb) of the chromosomes. The blue lines refer to low gene density, and the red lines refer to high gene density. (**B**) Collinearity analysis among *KH* family members in rice; the middle ring represents the twelve chromosomes and colored lines show the collinearity of *KH* family genes.

**Figure 2 ijms-25-05950-f002:**
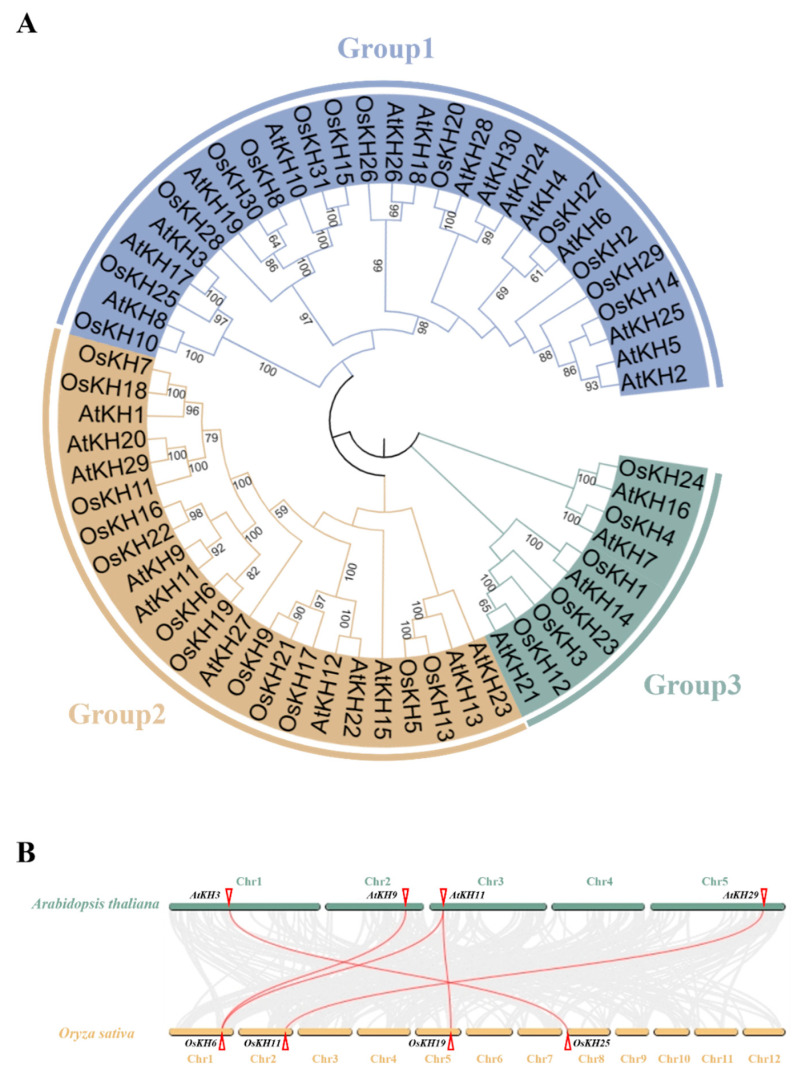
The phylogenetic tree and synteny analysis of 61 *KH* family genes between *Oryza sativa* and *Arabidopsis thaliana*. (**A**) Phylogenetic trees of the two species were constructed using MEGA7 by the neighbor-joining method with 1000 bootstrap replications; each cluster is labeled with different colors. (**B**) Synteny analysis of *KH* members between *Arabidopsis thaliana* and *Oryza sativa*; the gray lines represent all genes with synteny between the two species, and the red line marks the synteny of *KH* family genes.

**Figure 3 ijms-25-05950-f003:**
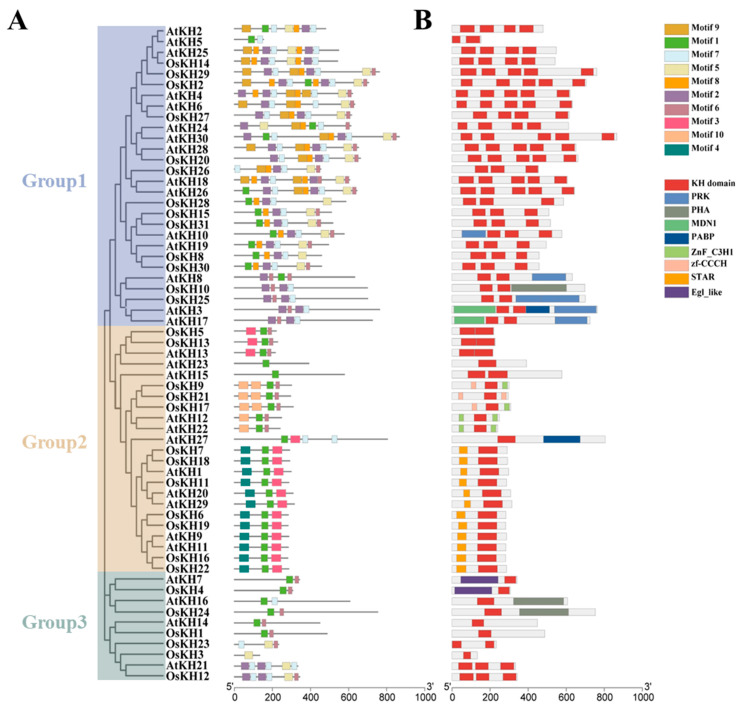
Conserved motifs and protein domains of 61 *KH* family members in *Oryza sativa* and *Arabidopsis thaliana*. (**A**) Conserved motifs of sixty-one *KH* family proteins in the two species were identified through the MEME online tool; ten conserved motifs are represented by different colors. (**B**) Nine conserved domains of *KH* family proteins were identified using the NCBI conserved domain database and are displayed in different colors.

**Figure 4 ijms-25-05950-f004:**
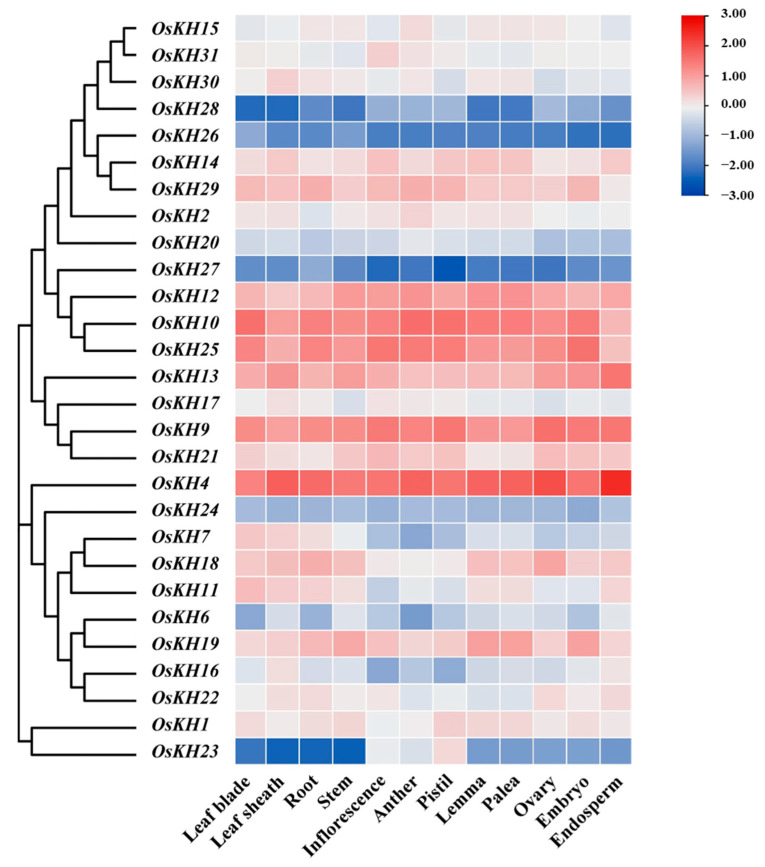
The expression pattern of *KH* genes in different tissues of rice. The expression analysis of *KH* family genes in twelve rice tissues; the blue and red squares relatively denote lower and higher expression levels. The data were downloaded from the rice public database (https://ricexpro.dna.affrc.go.jp/, accessed on 6 January 2024).

**Figure 5 ijms-25-05950-f005:**
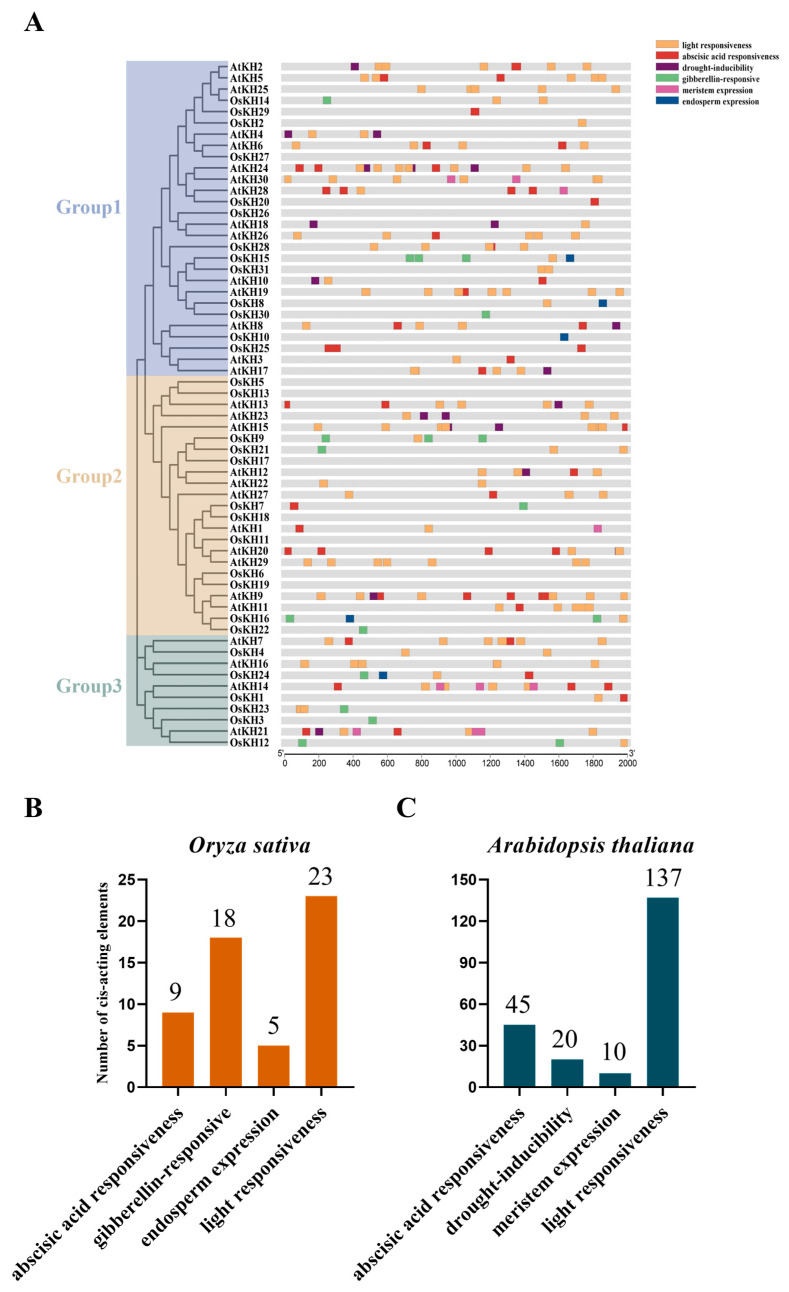
Analysis of cis-acting elements of *KH* family promoters in *Oryza sativa* and *Arabidopsis thaliana*. (**A**) Cis-acting elements analysis of sixty-one *KH* family gene promoters in the two species; the main six cis-acting elements were identified and are represented by different colors. (**B**) Quantitative statistics of cis-acting elements of *KH* family members in *Oryza sativa*. (**C**) Quantitative statistics of cis-acting elements of *KH* family members in *Arabidopsis thaliana*.

**Figure 6 ijms-25-05950-f006:**
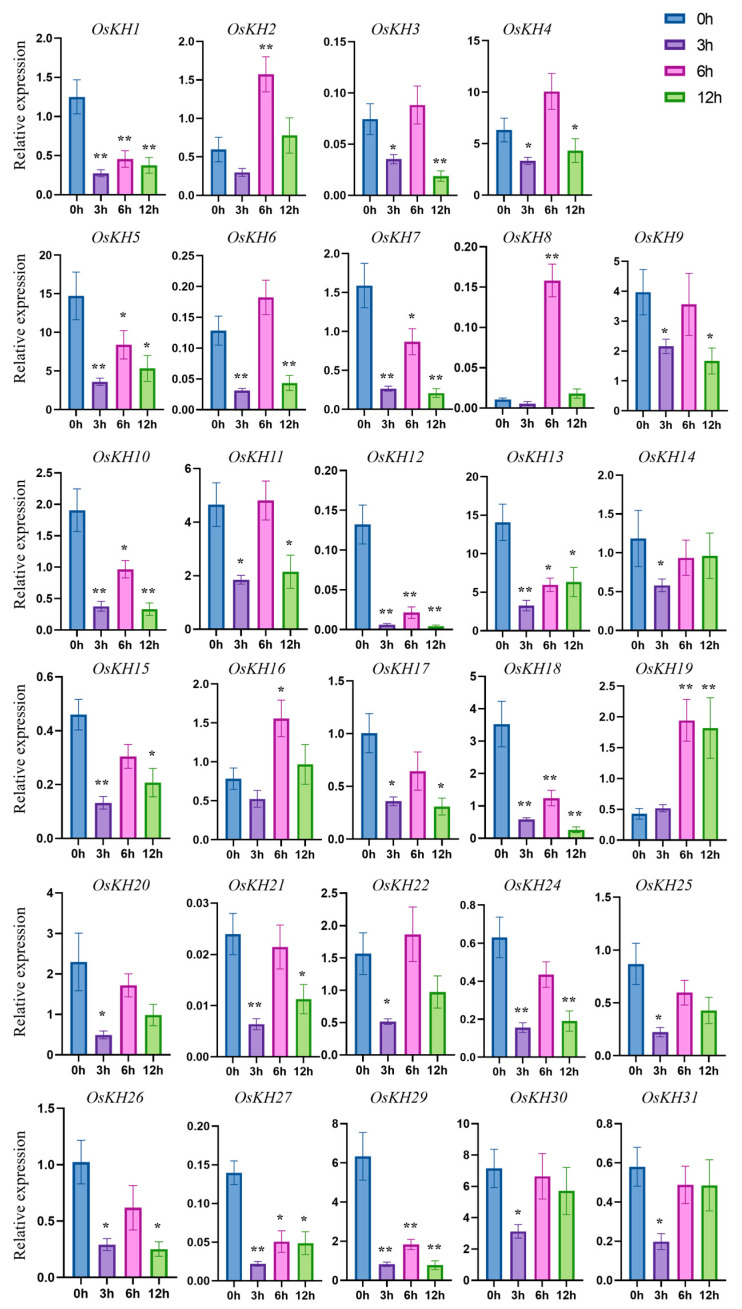
The expression analysis of *OsKH* genes in response to salt stress. The expression level of 31 *OsKH* genes was detected by qRT-PCR. The different colored columns represent different times of Nip seedling treatment. Values are presented as means ± SD (n = 3, * *p* < 0.05, ** *p* < 0.01; Student’s *t*-test).

**Figure 7 ijms-25-05950-f007:**
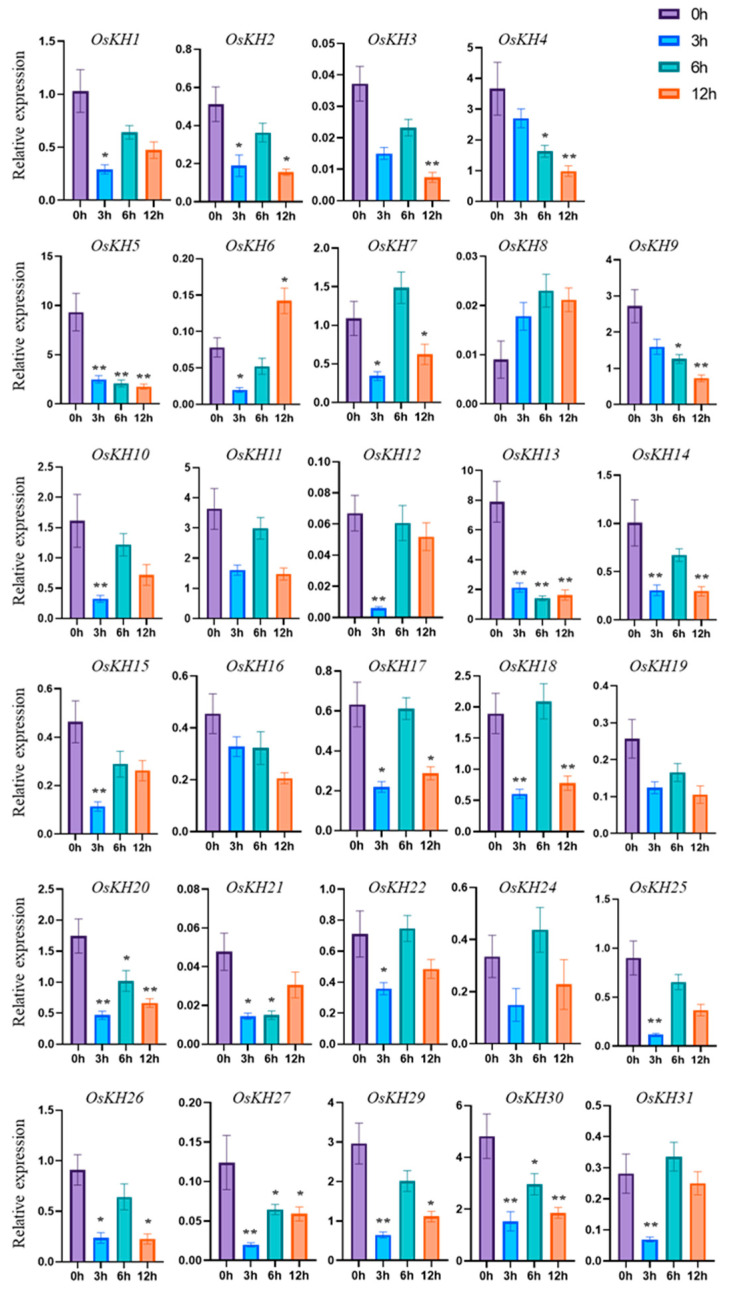
*OsKH* genes response to ABA in rice. Expression analysis of *OsKH* genes after the treatment of 50 µM ABA, the columns of different colors represent different times of Nip seedling treatment. Values are presented as mean ± SD (n = 3, * *p* < 0.05, ** *p* < 0.01, Student’s *t*-test).

**Figure 8 ijms-25-05950-f008:**
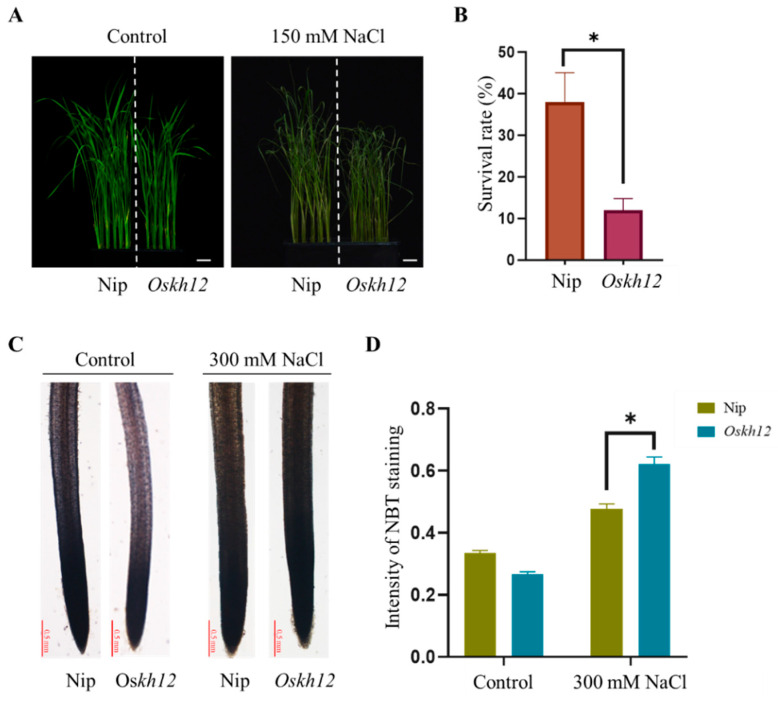
The *Oskh12* mutant displayed salt-sensitive phenotypes in rice. (**A**) The phenotypes of Nip and *Oskh12* under the treatment of 150 mM NaCl. Bars: 2 cm. (**B**) Survival rate statistics of Nip and *Oskh12* under the treatment of 150 mM NaCl. Values are presented as mean ± SD (n = 40, * *p* < 0.05, Student’s *t*-test). (**C**) ROS staining of Nip and *Oskh12* by NBT after the treatment of 300 mM NaCl. Bar, 0.5 mm. (**D**) NBT staining intensity of Nip and *Oskh12*; the data were calculated by ImageJ software 1.54g. Values are presented as mean ± SD (n = 5, * *p* < 0.05, Student’s *t*-test).

**Figure 9 ijms-25-05950-f009:**
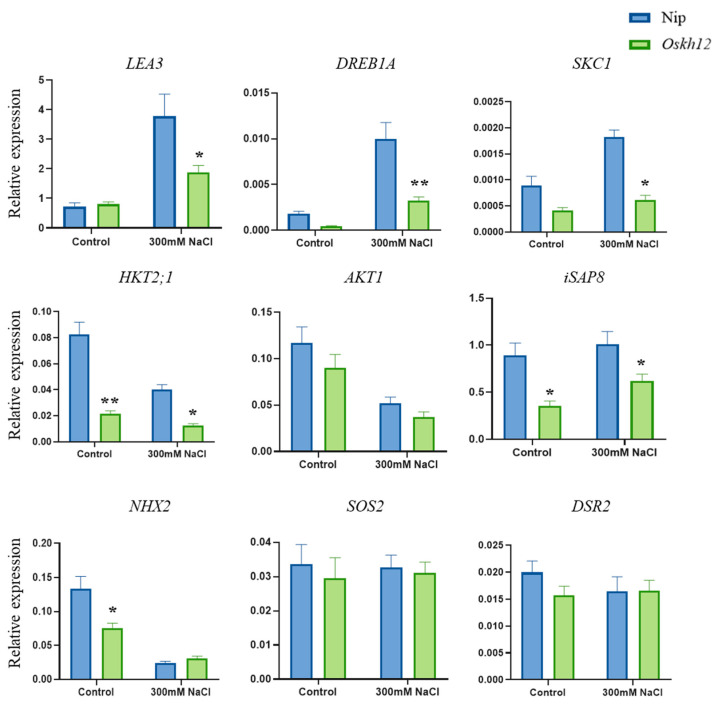
qRT-PCR analysis of salt-responsive genes in Nip and *Oskh12*. Expression analysis of 9 salt-responsive marker genes in Nip and *Oskh12* after the treatment of 300 mM NaCl. Values are presented as means ± SD (n = 3, * *p* < 0.05, ** *p* < 0.01; Student’s *t*-test).

**Figure 10 ijms-25-05950-f010:**
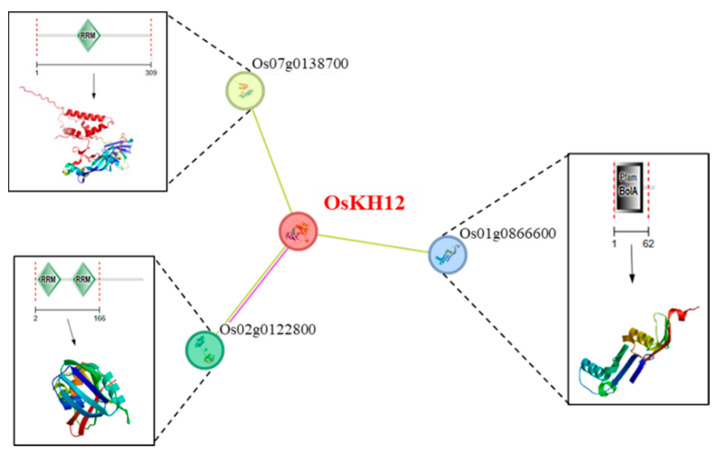
Functional regulatory network of OsKH12. Protein interactions of OsKH12 proteins were predicted using the STRING website. Red lines indicate protein fusion and yellow lines indicate text mining. The dotted line enlarges the structure of corresponding proteins.

**Table 1 ijms-25-05950-t001:** The characteristics of *KH* family proteins.

ID	Rename	Number of Amino Acids	MW (Da)	PI	Instability Index	Aliphatic Index	Grand Average of Hydropathicity	Location
At1g09660	*AtKH1*	298	33,687.72	8.42	61.41	75.87	−0.541	Nucleus
At1g14170	*AtKH2*	479	52,540.33	5.85	49.27	87.93	−0.361	Chloroplast
At1g33680	*AtKH3*	763	80,438.61	4.71	53.96	47.23	−0.96	Nucleus
At1g51580	*AtKH4*	621	67,115.01	5.74	50.65	75.17	−0.363	Chloroplast
At2g03110	*AtKH5*	153	17,376.29	4.16	34.48	90.39	−0.373	Peroxisome
At2g22600	*AtKH6*	632	67,754.64	5.94	37.66	96.33	−0.173	Chloroplast
At2g25910	*AtKH7*	342	38,474.69	5.53	49.65	94.09	−0.125	Mitochondria
At2g25970	*AtKH8*	632	64,606.8	5.32	57.97	40.32	−0.853	Nucleus
At2g38610	*AtKH9*	286	31,720.93	8.99	64.05	76.36	−0.56	Chloroplast
At3g04610	*AtKH10*	577	63,403.92	4.57	51.8	66.2	−0.753	Cytoplasm
At3g08620	*AtKH11*	283	31,500.94	9.3	59.94	79.93	−0.57	Nucleus
At3g12130	*AtKH12*	248	25,952.1	9.51	27.54	49.96	−0.623	Nucleus
At3g13230	*AtKH13*	215	24,009.02	9.71	27.82	88.93	−0.284	Cytoplasm
At3g16230	*AtKH14*	449	50,078.63	9.17	39.03	88.11	−0.275	Chloroplast
At3g29390	*AtKH15*	578	62,535.11	8.77	61.83	76.61	−0.51	Nucleus
At3g32940	*AtKH16*	607	66,479.76	9.01	47.82	69.11	−0.546	Nucleus
At4g10070	*AtKH17*	725	77,201.51	4.64	56.61	48.36	−0.939	Nucleus
At4g18375	*AtKH18*	606	65,760.82	8.04	50.66	87.94	−0.234	Chloroplast
At4g26000	*AtKH19*	495	54,024.67	4.92	51.46	86.28	−0.291	Cytoplasm
At4g26480	*AtKH20*	308	33,588.38	9.05	61.03	81.66	−0.391	Nucleus
At5g04430	*AtKH21*	334	36,018.23	5.77	50.4	76.86	−0.422	Nucleus
At5g06770	*AtKH22*	240	25,367.57	9.66	35.72	56.54	−0.628	Cytoplasm
At5g08420	*AtKH23*	391	45,323.08	9.53	57.5	64.83	−1.053	Nucleus
At5g09560	*AtKH24*	612	67,954	6.62	46.76	85.16	−0.35	Nucleus
At5g15270	*AtKH25*	548	59,712.05	8.33	58.14	80.24	−0.51	Nucleus
At5g46190	*AtKH26*	644	69,676.84	8.35	45.72	86.99	−0.283	Nucleus
At5g51300	*AtKH27*	804	87,050.31	5.88	55.54	55.11	−0.804	Nucleus
At5g53060	*AtKH28*	652	71,345	5.75	62.6	82.96	−0.478	Cytoplasm
At5g56140	*AtKH29*	315	33,984.66	7.87	62.06	80.48	−0.378	Nucleus
At5g64390	*AtKH30*	866	94,703.74	7.2	47.21	81.29	−0.395	Nucleus
Os01g0231900	*OsKH1*	487	54,345.92	8.42	48.93	80.1	−0.48	Chloroplast
Os01g0235800	*OsKH2*	705	76,040.62	5.96	36.44	83.21	−0.391	Mitochondria
Os01g0533450	*OsKH3*	133	14,786.94	10.07	31.48	95.26	−0.236	Chloroplast
Os01g0566900	*OsKH4*	307	34,744.41	5.25	43.35	95.21	−0.096	Cytoplasm
Os01g0788950	*OsKH5*	220	24,409.36	9.74	36.72	89.64	−0.198	Cytoskeleton
Os01g0818300	*OsKH6*	283	31,302.83	7.73	43.94	83.07	−0.451	Cytoplasm
Os01g0886300	*OsKH7*	290	32,556.19	8.77	54.67	84.31	−0.562	Nucleus
Os02g0125500	*OsKH8*	458	49,993.26	5.04	42.18	91.11	−0.352	Cytoplasm
Os02g0194200	*OsKH9*	300	31,153.44	9.58	35.7	48.23	−0.532	Nucleus
Os02g0224300	*OsKH10*	699	71,068.1	5.56	61.59	39.31	−0.887	Nucleus
Os02g0722700	*OsKH11*	286	31,368.76	9.27	46.16	83.25	−0.43	Chloroplast
Os02g0822300	*OsKH12*	343	36,718.26	6.19	53.38	76.82	−0.394	Cytoskeleton
Os03g0115200	*OsKH13*	227	24,965.89	9.74	40.16	86.43	−0.236	Cytoplasm
Os03g0376800	*OsKH14*	542	57,843.61	6.41	44.49	80.22	−0.395	Nucleus
Os03g0627500	*OsKH15*	510	54,819.89	4.81	53.11	65.43	−0.624	Nucleus
Os03g0815700	*OsKH16*	281	31,285.46	9.48	63.61	72.49	−0.617	Nucleus
Os04g0665700	*OsKH17*	309	31,787.8	9.48	41.21	47.9	−0.544	Cytoplasm
Os05g0419500	*OsKH18*	291	32,510.16	8.8	55.79	80	−0.53	Cytoplasm
Os05g0481500	*OsKH19*	282	31,240.65	8.65	63.45	74.04	−0.565	Nucleus
Os06g0342500	*OsKH20*	662	72,249.2	5.91	55.74	76.18	−0.577	Nucleus
Os06g0618100	*OsKH21*	295	30,451.57	9.53	38.8	49.46	−0.479	Nucleus
Os07g0227400	*OsKH22*	286	32,135.56	9.16	56.99	70.91	−0.685	Nucleus
Os07g0439100	*OsKH23*	233	24,955.9	4.61	28.88	79.53	−0.344	Nucleus
Os08g0100700	*OsKH24*	753	79,140.08	9.43	61.72	66.02	−0.504	Nucleus
Os08g0110800	*OsKH25*	700	72,186.11	4.88	60.17	41.61	−0.95	Nucleus
Os08g0200400	*OsKH26*	454	48,932.73	6.39	42.88	94.91	−0.207	Cytoplasm
Os09g0498600	*OsKH27*	616	65,484.48	7.31	49.72	72.44	−0.325	Mitochondria
Os10g0414700	*OsKH28*	586	65,147.67	5.68	59.13	80.24	−0.336	Cytoplasm
Os10g0495000	*OsKH29*	762	83,489.63	8.26	50.23	79.76	−0.594	Nucleus
Os10g0564000	*OsKH30*	458	48,410.67	5.21	43.83	90.37	−0.139	Cytoplasm
Os12g0597600	*OsKH31*	517	54,962.92	4.89	56.58	64.35	−0.66	Cytoplasm

## Data Availability

The sources of the data for this study are detailed in Section 4.

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
