# Peer review of "Identification and Functional Analysis of KH Family Genes Associated with Salt Stress in Rice"

_ijms, 2024, doi:10.3390/ijms25115950_

Round 1

Reviewer 1 Report

Comments and Suggestions for Authors

K-homologous (KH) family are nucleic acid-binding proteins containing KH domain with a role in splicing and transcriptional regulation.  In plants, KH domain family genes have been shown to take part in developmental processes, especially timing of flowering, and adaptation to environmental stresses. A systematic characterization of  the KH family in rice is lacking. In this respect, the manuscript contains new and valuable information. However, it is not convincingly written. The bioinformatics part is not well linked to the expression studies under salt stress, it seems that these are two artificially unified parts. The choice of salt stress is not well justified and properly described. The main responsive elements were identified in the promoter regions of KH genes, in rice - related primarily to light responsiveness, ABA (abscisic acid), and less – to GA  and endosperm. The link between ABA and salt stress should be better explained in the introduction in order to better justify the use of salt stress as a treatment.  Besides, it is not clear for me why the authors have chosen to knockout the OsKH12 gene which exhibited the strongest decrease in expression level during salt treatment, not an increase, and claim that Oskh12 acts as a positive regulator in salt-stress responses. The observed reduction in the expression of several key salt-tolerant genes in the knockout Oskh12 mutant may have other reasons than their regulation by OsKH12 gene. The authors have established that OsKH genes are mostly constitutively expressed.  In my opinion, the manuscript should be improved to gain more consistency.

Some remarks:

Abstract line 15 “in responding to abiotic stress” – unclear, could be omitted. Line 22 “Oskh12 acts as a positive regulator involved in salt-stress responses” – not convincing

Introduction: line 42 – adaptation to environmental stresses; instead of aim of the study there is a short repetition of the main results. The ABA role in salt  stress should be briefly explained.

MMs line 138 – “The japonica varieties Nipponbare” –actually only one variety is used. Lines 141-142 – “Two-week-old seedlings of rice were separately processed with 150 mM NaCl, 300 mM NaCl, and 50 μM ABA” – how it was applied – into the medium? For how long? Line 146 – RNA extraction from whole plants, roots or leaves?

Results – line 166 “chromosome 1 has the most seven…” unclear; line 182 – “amino acid numbers” – number of amino acids; line 193,195, table 1 – chloroplastid – did you mean chloroplast? Line 244 – leaf blade. In my opinion, the transcript response to ABA should be moved from supplementary into the main text. It is not clear for me why transcript abundance was analysed at 300 mM NaCl treatments whereas the phenotypic stress response of Nip and homozygous Oskh12 mutant - at 150 mM NaCl treatment – what was the reason? Fig 7 - ROS staining of Nip and kh12 by NBT after the treatment of 300 mM NaCl – in the controls ROS production in the mutant is lower which could mean a general slowdown in the metabolism and not a specific salt stress effect. Some of the salt responsive genes are also lower in expression in the control mutant plants.

Discussion – Reading this ms I am not convinced that OsKH12 function as a positive factor in response to salt stress by regulating the expression of salt-responsive  genes, in my opinion this is rather speculative statement. The primary role of OsKH genes should be in plant growth and development, as it could be seen from promotor analysis, and in the second place – in stress response and probably stress tolerance.  The possible interaction proteins of OsKH12 protein with proteins containing the RRM 358 domain (Figure S4) should not be in the Discussion section but in the Results.

Comments on the Quality of English Language

completely understandable

Author Response

Thank you very much for taking the time to review this manuscript. Please see the response in the attachment. 

Reviewer 2 Report

Comments and Suggestions for Authors

The manuscript entitled “Identification and functional analysis of KH family genes associated with salt stress in rice” authored by Xie etal    constitutes a in silico-based identification of KH proteins and experimental validation of function of one of them.

Overall, the study is well designed, however, there are some issues, that make it inappropriate to be published as it is now.

1.     Apart from Oskh12, another low responsive to salt stress, KH gene should have been knocked out in order to compare all downstream steps of the present research. Please explain why such experiments have not been performed or how can the authors be sure that their results are adequately controlled.

2.     Although the strategy of sequence retrieval is described in Material and methods, a short reference of the name of the source should be given in the beginning of 3.1, 3.2, 3.3, 3.4, 3.5 paragraphs.

3.     Paragraph 3.4: it should be stressed that results are obtained from databases.

4.     It is of absolutely need to report (3.6 paragraph) the tissue that has been used to obtain the results described in figure 6 and supplementary figure 3. In addition, these results should be compared with results from in silico retrieved expression levels shown in figure 4.

5.     Figure 3: It should be explained how these motifs were constructed. How were the limits decided upon (number of aa)? Please give a rough correspondence between motifs and domains. If this cannot be done, please explain why. It seems that KH domain does not have a consistent/conserved sequence. Please elaborate appropriately.

6.     Figure 5b: for reasons of comparison of conservation it would be better if two diagrams were presented, one for each plant.

7.     Line304: please elaborate (or give REF) how these genes are related to salt stress.

8.     Line 311 and 343: I, personally, can see that oskh12 is involved in the regulation of salt response, however, not in a positive manner. If authors think so, they should present more evidence.

Minor points

Line 22 and 23: oskh12 is used as both gene name and plant having this gene nocked out. Please amend and keep a consistent nomenclature.

Line 17: and everywhere in the text that is referred to results from investigations in database (PSORT base): please write “in silico”, otherwise it is misleading.

Lines 15 and 17: It is not clear for the 31, 30 and 61 genes, the organisms and the experiments they refer to.

Lines 77-89: It is a summary of the findings; however, the aim of the study should be highlighted in this last paragraph of introduction.

Line 98: how CDSs were extracted and from which datbases?

Line 111: which set, please explain.

Line 139: please explain give ref on the Nip wd plant.

Line 153: Please elaborate what does ROS-NBT assay measures and how does this corelates with salt -stress response.

Line 193: Please explain what does ‘small partial proteins’ mean.

Tabe 1 legend: Please denote that these characteristics come from PSORT database coming from experimental data or are predicted from the tool PSORT.

Line 204: An alternative hypothesis may exist i.e. come from a common ancestor. So, the authors should be more cautious with their conclusions.

Line 251-252: too general for results section. It should be more specific.

Line 264: Endemic in not the proper word. Please amend.

263-269: not understandable

262-263: please explain how these pathways emerged.  

Figure 7: confusing labeling. Please correct.

Line 300: please explain here or elsewhere appropriately, how is ROS production and NBT staining is related to salt-stress response.

Syntax errors in lines:

153, 166, 183-188, 203, 209,211, 227, 230, 303, 296, 334-336, 339, 340

Comments on the Quality of English Language

Minor editing of English language required

Author Response

(The authors gave the same response as above.)

Round 2

Reviewer 2 Report

Comments and Suggestions for Authors

The manuscript has been substantially improved.

However, before it is accepted for publication paragraph 3.8 has to be reformulated and the title should be changed. As the authors say in legend of figure 10, the connections represent either data from fusion experiments or text mining. Thus, there is no experimental prof that these proteins do interact physically. They may interact and this has to be tested experimentally. So, the authors should be more cautious with what they claim. Title should be something like

OsKH12 "MAY INTERACT WITH" or "IS RELATED TO" proteins containing the RRM domain , or something similar

Minor points

Line 234 change to “conserved.”

Line 278 change to “were”

Line 385   please, do not start a sentence with “And”

Author Response

Thank you for your careful review of this manuscript. Following your suggestion,  the manuscript has been revised. Please see the attachment. 
